Journal of
open psychology data

# Data from ManyDogs 1

DATA PAPER

MANYDOGS PROJECT

JULIA ESPINOSA ⓘ

ELIZABETH HARE ⓘ

DANIELA ALBERGHINA ⓘ

BRYAN MITCHEL PEREZ VALVERDE ⓘ

JEFFREY R. STEVENS ⓘ

*Author affiliations can be found in the back matter of this article

]u[ ubiquity press

## ABSTRACT

The ManyDogs 1 study is the first multi-site collaborative study of dogs' responses to human pointing. It addressed whether dogs perceive the gesture as socially communicative and are therefore more likely to follow the point when it is paired with additional social signals (ManyDogs Project, et al., 2023b). Researchers from 20 research sites across eight countries collected data from 704 dogs. Here, we present not only the behavior data on the dogs' responses to experimental pointing conditions but also guardian responses to survey questions, including the Canine Behavior and Research Questionnaire (C-BARQ, Hsu and Serpell, 2003). This dataset allows for assessing associations among C-BARQ measures as well as connections to the experimental task data, research site metadata, and other dog and guardian characteristic data.

CORRESPONDING AUTHOR:
**Jeffrey R. Stevens**

B83 East Stadium, University of Nebraska-Lincoln, Lincoln, Nebraska 68588, USA

jeffrey.r.stevens@gmail.com

KEYWORDS:
Canine; Dog; Interspecies interaction; Pointing; Social communication

TO CITE THIS ARTICLE:

# (1) BACKGROUND

ManyDogs is an international research consortium of scientists with a shared interest in the factors driving canine behavior and cognition (ManyDogs Project et al., 2023a). This consortium actively fosters a diverse community and formalizes a transparent and equitable process for engaging in multi-site collaborative projects related to canine behavior and cognition. In the first ManyDogs study—named ManyDogs 1 (ManyDogs Project et al., 2023b)—we investigated a question of theoretical importance in canine science: Do dogs act on human pointing signals as though they are communicative social cues? Domestic dogs (*Canis familiaris*) have become a popular animal model for investigating behavioral and cognitive evolution due to their shared ecological niche with humans and because they are plentiful, easy-to-access research subjects in many parts of the world. Unlike humans' more closely related primate relatives (e.g., chimpanzees, *Pan troglodytes*) and laboratory-bred rodent models of behavior and cognition, dogs are embedded in the human environment, living in our homes and navigating our workplaces. Dogs have been intentionally bred to live in these spaces and interact with humans, making them a ready comparison species in which to investigate the origins of cognitive processes. Interest in their putatively innate ability to interact and cooperate with humans has made them particularly popular in comparative studies, especially as they appear to respond to human communicative cues—such as pointing—more accurately and flexibly than other species (e.g., Bräuer et al., 2006). Though point following behavior in dogs has been widely observed and studied over recent decades (Miklósi et al., 1998; Soproni et al., 2001; Hare et al., 2002; Kaminski & Nitzschner, 2013), there is still disagreement as to the underlying motivation for the behavior. Do dogs respond to pointing because they interpret the gesture as socially communicative (Hare & Tomasello, 1999; Soproni et al., 2001; Kaminski & Nitzschner, 2013)? Or rather, because dogs have learned to associate human pointing with food rewards (e.g., Wynne et al., 2008)?

To investigate this question, we used a big team science, single-study approach, modeled after other groups such as ManyBabies (Frank et al., 2017) and ManyPrimates (ManyPrimates et al., 2019). Big team science involves "endeavors in which an unusually large number of researchers — often dispersed across institutions and world regions — self-organize to pool intellectual and material resources in pursuit of a common goal" (Coles et al., 2022). With this approach, multiple research teams followed the same experimental protocol, sharing the high cost of behavioral data collection and striving to implement the method in an identical manner. This approach replicated the study simultaneously in different research environments and with different populations.

Big team science is important in animal cognition work generally because it greatly increases sample sizes and diversity and enhances task design (Alessandroni et al., 2024). This approach is particularly important in canine cognition because, due to the larger and more diverse samples, big team science allows us to answer new questions previously unattainable with smaller, more homogeneous samples (ManyDogs Project et al., 2023a). This includes the role of breed, life history, training, and geographical location on behavior.

Under our main hypothesis, we predicted that when dogs saw a pointing gesture paired with *ostensive* signals, such as dog-oriented eye gaze and dog-directed speech (i.e., calling the dog's name), they would be more likely to follow the gesture than when no such ostensive cues accompanied the point. If we observed this response across dogs, the result would lend support to the idea that explicitly communicative cues help dogs understand the intention behind the gesture. Such an outcome would suggest that dogs find ostensive cues necessary for understanding pointing, similar to human children (Behne et al., 2005). On the other hand, if no difference was observed in point following across the ostensive and non-ostensive conditions (pointing without additional voice or gaze cues), this outcome would suggest that dogs indiscriminately follow pointing. Such a result would suggest that dogs raised by humans may learn to associate pointing limbs with rewards and not necessarily perceive any communicative intention underlying the gesture.

In addition to testing our main hypothesis, we took the opportunity offered by multiple research teams in different sites collaborating on the same study to collect data on sources of inter-site variability that could influence the results. Often, studies by different groups produce inconsistent results (Rodriguez et al., 2021). The impact of cultural differences in scientific practice, dog training norms across regions, and of course variation in heritable traits across dog breeds have complicated replication studies conducted by isolated groups, making it difficult to pinpoint the reasons for inconsistent results. By collecting extensive and detailed information about the testing environments and subject population, we achieved a rich and robust dataset that would support investigation about multiple influences on dogs' behavior previously out of reach.

# (2) METHODS

## 2.1 STUDY DESIGN

The ManyDogs 1 study used a cross-sectional, multi-method approach to collecting data. Dog guardians were recruited through the individual research sites' existing databases and via their respective outreach methods (e.g., social media). Prior to participating in the behavioral tasks at a research site, guardians completed an online

survey, providing basic environment and demographic information along with a validated assessment of canine temperament and behavior—the Canine Behavioral Assessment and Research Questionnaire (C-BARQ, Hsu & Serpell, 2003). The behavioral tasks included a short series of object-choice warm-ups that acclimated the dog to the space, followed by two experimental pointing conditions. Using a within-subjects design, dogs were tested on two different pointing cues by a trained researcher, ostensive and non-ostensive, in counterbalanced orders across subjects. Response rates to these two styles of pointing were compared within subjects, while additional between-subject variables derived from the survey data supported investigating variability in behavior as a function of demographic and environmental factors.

## 2.2 TIME OF DATA COLLECTION

Data for the study were collected over 13 months, between January 2022 and January 2023. Within this time window, research sites were able to decide when to implement the protocol according to the guardian and staff availability (collection dates available in dataset).

## 2.3 LOCATION OF DATA COLLECTION

For the main study, data were collected in 20 research sites across eight countries (Argentina, Canada, Croatia, Hungary, Italy, Poland, UK, USA) on three continents (Figure 1). In addition, an Austrian site recorded only pilot data and is not represented in this dataset. A full list and description of research sites is available in Table 1.

## 2.4 SAMPLING, SAMPLE AND DATA COLLECTION

Across all sites, teams behaviorally tested 704 dogs (M:F = 334:373, mean ± SD age = 4.40 ± 3.1 years [range = 0.3–20.8]). Approximately 76.9% of the dogs were spayed or neutered, 53.8% were of single-breed ancestry (comprising 85 distinct breeds), 90.2% lived in private homes, 9.6% lived in group/kennel housing, and 0.3% lived in other housing. We excluded 235 dogs because they did not complete the behavioral testing and 14 dogs because of experimenter errors. Thus, complete behavioral data were collected from 455 dogs, and complete survey data were collected from 495 dogs. Guardians identified as female (81.0%), male (17.7%), and nonbinary/other (1.3%) with a modal guardian age range of 30–39 years. All labs that started data collection met our criteria for inclusion, so no labs were excluded.

## 2.5 MATERIALS/SURVEY INSTRUMENTS

The guardian survey was hosted on Qualtrics (complete survey available at https://doi.org/10.17605/osf.io/7rwpc). The survey included dog demographics (name, living situation, sex, neuter status, birth date, breed information, acquisition type), training information (communication style and frequency, training experience, research experience), guardian demographics (gender, age, community type), and C-BARQ. The C-BARQ trainability scale (eight items) was presented first and was included in the pre-registered analysis of pointing (ManyDogs Project et al., 2023b). After answering the trainability questions, guardians could decide to submit their responses or continue to complete the remaining six behavior assessment scales from the C-BARQ. If they continued, they answered questions about aggression (28 questions), fear (18 questions), separation-related behavior (9 questions), excitability (7 questions), attachment/attention-seeking (7 questions), and miscellaneous behavior problems (28 questions), including chasing, chewing, begging, pulling, urinating,

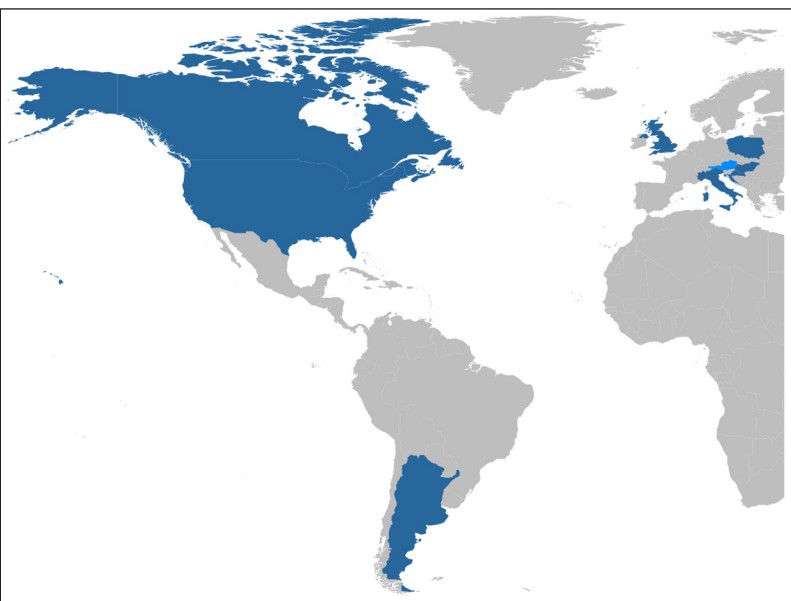

**Figure 1** ManyDogs 1 data presented here were collected from 20 research sites in eight countries: Argentina, Canada, Croatia, Hungary, Italy, Poland, UK, USA (dark blue). Pilot data not included in this dataset were collected from a site in Austria (light blue).

| SITE | LOCATION | DATA ABBREVIATION |
|------|----------|-------------------|
| Animal Health and Welfare Research Centre | Winchester, United Kingdom | ucs |
| Arizona Canine Cognition Center | Tuscon, AZ, USA | accc |
| Auburn Canine Performance Sciences | Auburn, AL, USA | auburn |
| Boston Canine Cognition Center | Boston, MA, USA | bccc |
| Brown Dog Lab | Providence, RI, USA | bdl |
| Canid Behavior Research Group | Buenos Aires, Argentina | icoc |
| Canine Cognition and Human Interaction Lab | Lincoln, NE, USA | cchil |
| Canine Cognition Center at Yale | New Haven, CT, USA | yale |
| Canine Companions | Santa Rosa, CA, USA | cci |
| Canine Research Unit | St. John's, NL, Canada | crumun |
| Clever Dog Lab* | Vienna, Austria | cdl |
| Comparative Cognition Lab | Winnipeg, MB, Canada | manitoba |
| Comparative Cognitive Science Lab | Rijeka, Croatia | urijeka |
| Consultorio Comportamentale | Messina, Italy | umessina |
| Department of Psychology and Individual Differences | Warsaw, Poland | uwarsaw |
| Dog Cognition Centre | Portsmouth, United Kingdom | dcc |
| Duke Canine Cognition Center | Durham, NC, USA | duke |
| Leader Dogs for the Blind | Rochester, MI, USA | ldbtdc |
| Social Cognition Lab | Dundalk, ON, Canada | queensu |
| The Family Dog Project | Budapest, Hungary | eltebuda |
| Thinking Dog Center | New York City, NY, USA | tdc |

**Table 1** Site information.

*Clever Dog Lab participated only in the pilot data collection.

defecating, barking, and licking. Most questions used a 5-point Likert scale with a Not Observed option. Some categories included open-ended questions for additional explanations of their dog's behavior, but we did not include them in our dataset to protect guardian anonymity.

To facilitate replication of the methodology, the detailed experimental protocol is open-access and available with the original scientific report (ManyDogs Project et al., 2023b). Behavioral data were collected at individual research sites, where guardians brought the dogs in for test sessions. The study was designed to take 30 minutes or less and had two stages, warm-ups and test trials. After the dogs acclimated to the testing room, they participated in a series of warm-up object-choice tasks. The first task piqued the dogs' interest in food rewards and gauged their willingness to approach the experimenter and pick up visible food from the floor. Each dog completed two visible food placement trials. The second task built up an association between cups and food. In this task dogs completed a minimum of three trials in rapid succession without being recalled to the start line. There were no performance

requirements for the first two warm ups, only that the dog should retrieve food and make contact with the cups, showing a willingness to engage in the task and approach the experimenter. The third and fourth warm up tasks scaffolded the more formal trial structure and familiarized the dog with the two lateral search locations on either side of the experimenter. The third task used one cup with visible baiting at each of the lateral search locations and dogs completed four trials, two per side in alternating order. The fourth and final warm up used two cups and the same visible baiting procedure as in one-cup warm-up. In two-cup warm-up, dogs had to choose the visibly baited cup over the empty cup on four out of six consecutive trials in a sliding window of opportunity to progress to the test trials. A maximum of 20 two-cup trials were allowed. All warm-up tasks required two individuals: an experimenter to bait and place the cups and a handler to release the dog to make a choice and recall for subsequent trials (handlers could be either trained researchers or the dog's guardian).

Once meeting the two-cup warm-up task criteria, the dogs moved on to two experimental conditions and were required to complete eight trials per condition (condition

order counterbalanced between subjects). In the non-ostensive condition, the experimenter looked at the floor and cleared their throat while holding a piece of food in front of their body for the dog to notice before placing the food underneath one of two cups behind a visual occluder. They then removed the occluder and moved each of the cups to one of the lateral search locations. When the cups were in place, the experimenter cleared their throat and made a contralateral momentary point to the baited cup, holding the gesture for 2 seconds before retracing their hand and the handler released the dog to make a choice. The ostensive condition used the exact same baiting procedure and pointing gesture, but instead of clearing their throat and looking down, the experimenter used two ostensive cues to get the dogs' attention. These cues, dog-directed speech and dog-directed gaze, were modeled on previous work where dogs had followed intentional, direct cues from the experimenter (Miklósi et al., 1998; Soproni et al., 2001; Hare et al., 2002; Kaminski & Nitzschner, 2013; Tauzin et al., 2015). The vocal cue the experimenter gave was "[dog name], look!", and they gazed at the dog while showing the food and giving the pointing gesture. The two test conditions were separated by a one-minute play break and re-familiarization with the testing situation. After the two experimental conditions, the dogs completed an odor control condition with a similar set-up as the ostensive condition, except no point cue was given. The control was intended to determine whether the dogs were using olfactory instead of visual cues to solve the task.

## 2.6 QUALITY CONTROL

Collecting high-quality data was a key objective of ManyDogs 1. To validate the study design and analysis plan, we conducted a pilot experiment at a single site with 91 dogs. We pre-registered the pilot study at the Open Science Framework (https://osf.io/gz5pj/). The pilot data are not included in this dataset.

For the primary study presented here, we pre-registered the hypotheses, methods, and analysis plan as a registered report at *Animal Behavior and Cognition* (https://doi.org/10.31234/osf.io/f86jq). Because this study involved multiple sites running the same protocol, we sought to ensure consistent implementation across sites. During a researcher training phase, participating sites were required to submit videos of their team performing the protocol, as well as the full set of videos from the first dog tested. Two project administrators reviewed the videos for all sites and provided feedback on each site's implementation to improve consistency across sites.

Behavioral tests were video recorded and experimenters also live-coded the dog's responses on paper. Data were compiled across sites through a data entry survey hosted on Qualtrics. Using a survey protected the resulting data file from errors associated with multiple individuals directly editing the file. To measure inter-rater reliability of the live coding of experimental sessions, each site had a research assistant blind to the project's focus recode a subset of sessions. If a site did not have an assistant blind to the study, an assistant from another site coded the videos. This recoding resulted in an overall Cohen's kappa of 0.98 with individual sites ranging from kappa = 0.92–1.00.

## 2.7 DATA ANONYMIZATION AND ETHICAL ISSUES

Each research site participating in this study obtained approval from their respective institutional ethics committee (see Table S1 of ManyDogs Project et al., 2023b). All guardians provided informed consent to participate and were free to discontinue from the study at any time.

All identifiable information has been removed from the dataset, including replacing dog names with ID numbers.

## 2.8 EXISTING USE OF DATA

The behavioral data and a portion of the guardian data collected for the ManyDogs 1 study was used and published in:

ManyDogs Project, Espinosa, J., Stevens, J.R., Alberghina, D., Barela, J., Bogese, M., Bray, E., Buchsbaum, D., Byosiere, S.-E., Cavalli, C., Dror, S., Fitzpatrick, H., Freeman, M.S., Frinton, S., Gnanadesikan, G., Guran, C.-N.A., Glover, M., Hare, B., Hare, E., Hickey, M., Horschler, D., Huber, L., Jim, H.-L., Johnston, A., Kaminski, J., Kelly, D., Kuhlmeier, V.A., Lassiter, L., MacLean, E., Ostojic, L., Pelgrim, M.H., Pellowe, S., Salomons, H., Santos, L., Silver, Z.A., Silverman, J.M., Sommese, A., Völter, C., Walsh, C., Worth, Y.A., Zipperling, L.M.I., Żołędziewska, B., and Zylberfuden, S. G. (2023). ManyDogs 1: A multi-lab replication study of dogs' pointing comprehension. *Animal Behavior and Cognition*, 10(3), 232–286. https://doi.org/10.26451/abc.10.03.03.2023

# (3) DATASET DESCRIPTION AND ACCESS

The dataset contains 704 observations of 210 variables described in a codebook and Table 2. The dataset contains variables supplied by a survey as well as experimental variables. Data provided by each dog's guardian include demographic information about the dog and guardian, responses to questions about the types and frequencies of the dog's training activities, and answers to the C-BARQ.

In addition to the data provided by guardians, experimental variables are included in this dataset. These include information about whether the dog completed the experiment and was used in the analysis, experimental conditions, and trial-by-trial data on correct choices (choosing the cup baited with a treat).

| CATEGORY OF VARIABLE | VARIABLE NAME | DESCRIPTION | QUESTION TEXT | POSSIBLE RESPONSE VALUES |
|---|---|---|---|---|
| Dog Demographics | date | Date | Timestamp for completion of questionnaire | YYYY-MM-DD |
| | site | Abbreviation for site/location | What location are you going to visit? | accc, auburn, bccc, bdl, cchil, cci, crumun, dcc, duke, eltebuda, icoc, ldbtdc, manitoba, other, queensu, tdc, ucs, umessina, urijeka, uwarsaw, yale |
| | subject_id | Subject ID | What is your dog's assigned subject ID? | [site abbreviation]_[study number]_[subject number] |
| | experiment_status | Status of subject in experiment | | Error (Experimental error invalidated session), Incomplete (Subject did not complete session, invalidating it), Included (Valid session used in analysis) |
| | owned_status | Location of where dog lives | What is the dog's living situation? – Selected Choice | Group housing (e.g., working dog kennel), Private home, Other |
| | birthdate | Dog date of birth | Dog date of birth | YYYY-MM-DD |
| | sex | Dog sex | What is your dog's sex? | Female, Male |
| | age | Dog age (years) | Dog age in years | Number |
| | desexed | Dog neuter status | Has your dog been spayed or neutered? | Yes, No |
| | purebred | Dog purebreed status | Is your dog purebred? | Yes, No |
| | breed | Dog breed | What breed is your dog? | Multiple choice; 95 breeds represented |
| | breed_registry | Dog breed registration status | Is your dog registered with a kennel club in your country? | Yes, No |
| | mixed_breed | Dog mixed breeds | Is your dog a mix of known breeds? | Yes, No |
| Training and Communication | communication_method | Owner's method of communication with dog | How do you typically communicate with your dog? Select all that apply | Acoustic (clicker or whistle), Gesture (hand gestures, pointing), Verbal (spoken words), Other |
| | gesture_frequency | Frequency of owner using hand gestures with dog | How frequently do you use hand gestures (such as pointing or waving) to communicate with your dog? | Never, Seldom, Sometimes, Usually, Always, Not observed |
| | gaze_follow | Frequency of dog following pointing gestures | My dog follows pointing gestures with it's gaze immediately | Never, Seldom, Sometimes, Usually, Always, Not observed |
| | training_type | Presence of dog training/activities | Indicate the frequency with which your dog has participated in each of the following types of training/activity in the past 12 months. Select all that apply. | Puppy (Puppy class), Neighbor (Good neighbor class), Obedience1 (Basic obedience), Obedience2 (Advanced obedience), Rallyo (Rally obedience), Music (Musical freestyle), Agility, Ballsport (flyball), Discdog, Conform (Conformation), Scent, Search_rescue (Search and rescue), Sled (Sled pulling/cart pulling), Pullsport (Skijoring/Canicross/Bikejoring), Therapy, Service, Hunt (Game hunting/tracking), Herd (Herding/sheepdog trials), Other |
| | training_freq_puppy | Frequency of puppy classes | Puppy class frequency of participation in the last 12 months | Never, Weekly, >1 week, <1 month, 1–2 month |

(Contd.)

| CATEGORY OF VARIABLE | VARIABLE NAME | DESCRIPTION | QUESTION TEXT | POSSIBLE RESPONSE VALUES |
|---|---|---|---|---|
| | `training_freq_ neighbor` | Frequency of good neighbor classes | Good neighbor class frequency of participation in the last 12 months | Never, Weekly, >1 week, <1 month, 1–2 month |
| | `training_freq_ obedience1` | Frequency of basic obedience classes | Basic obedience frequency of participation in the last 12 months | Never, Weekly, >1 week, <1 month, 1–2 month |
| | `training_freq_ obedience2` | Frequency of advanced obedience classes | Advanced obedience frequency of participation in the last 12 months | Never, Weekly, >1 week, <1 month, 1–2 month |
| | `training_freq_ rallyo` | Frequency of rally obedience activities | Rally obedience frequency of participation in the last 12 months | Never, Weekly, >1 week, <1 month, 1–2 month |
| | `training_freq_music` | Frequency of musical freestyle activities | Musical freestyle frequency of participation in the last 12 months | Never, Weekly, >1 week, <1 month, 1–2 month |
| | `training_freq_ agility` | Frequency of agility activities | Agility frequency of participation in the last 12 months | Never, Weekly, >1 week, <1 month, 1–2 month |
| | `training_freq_ flyball` | Frequency of flyball activities | Flyball frequency of participation in the last 12 months | Never, Weekly, >1 week, <1 month, 1–2 month |
| | `training_freq_disc` | Frequency of discdog activities | DiscDog frequency of participation in the last 12 months | Never, Weekly, >1 week, <1 month, 1–2 month |
| | `training_freq_ conform` | Frequency of conformation activities | Conformation frequency of participation in the last 12 months | Never, Weekly, >1 week, <1 month, 1–2 month |
| | `training_freq_scent` | Frequency of scent detection activities | Scent detection frequency of participation in the last 12 months | Never, Weekly, >1 week, <1 month, 1–2 month |
| | `training_freq_ search` | Frequency of search and rescue activities | Search and rescue frequency of participation in the last 12 months | Never, Weekly, >1 week, <1 month, 1–2 month |
| | `training_freq_sled` | Frequency of sled pulling activities | Sled pulling/cart pulling frequency of participation in the last 12 months | Never, Weekly, >1 week, <1 month, 1–2 month |
| | `training_freq_ pullsport` | Frequency of skijoring/canicross/ bikejoring activities (attaching dogs to skiers/runners/bikers) | Skijoring/Canicross/Bikejoring frequency of participation in the last 12 months | Never, Weekly, >1 week, <1 month, 1–2 month |
| | `training_freq_ therapy` | Frequency of therapy dog activities | Therapy/ambulance dog frequency of participation in the last 12 months | Never, Weekly, >1 week, <1 month, 1–2 month |
| | `training_freq_ service` | Frequency of service dog activities | Specialized service training frequency of participation in the last 12 months | Never, Weekly, >1 week, <1 month, 1–2 month |
| | `training_freq_hunt` | Frequency of hunting/tracking activities | Game hunting/tracking frequency of participation in the last 12 months | Never, Weekly, >1 week, <1 month, 1–2 month |

| CATEGORY OF VARIABLE | VARIABLE NAME | DESCRIPTION | QUESTION TEXT | POSSIBLE RESPONSE VALUES |
|---|---|---|---|---|
| | training_freq_herd | Frequency of herding activities | Herding/sheepdog trials frequency of participation in the last 12 months | Never, Weekly, >1 week, <1 month, 1–2 month |
| | training_freq_ other1 | Frequency of other activities (fill in activity) | Other frequency of participation in the last 12 months (1) | Never, Weekly, >1 week, <1 month, 1–2 month |
| | training_freq_ other2 | Frequency of other activities (fill in activity) | Other frequency of participation in the last 12 months (2) | Never, Weekly, >1 week, <1 month, 1–2 month |
| | training_freq_ other3 | Frequency of other activities (fill in activity) | Other frequency of participation in the last 12 months (3) | Never, Weekly, >1 week, <1 month, 1–2 month |
| | lab_exposure | Research lab experience status | Has your dog participated in research studies before at this or another location/institution? | Yes, same site; Yes, different site; No; Unsure |
| | research_experience | Research lab experience types | What type of research tasks has your dog participated in during previous visits to research centers? | Choice tasks, Cup tasks, Human point, Other |
| | other_household_ dogs | Dog shared household status | Does your dog currently live with other dogs? | Yes, No |
| | num_household_dogs | Number of dogs in household | If yes, how many? | Number |
| Guardian Demographics | years_owned | Number of years dog lived with owner | Approximately, how many years have you owned your dog? | Number |
| | origin | Dog origin | How did you acquire your dog? | Breeder, Relation, Rescue, Shelter, Other |
| | guardian_gender | Guardian gender | With which gender do you most identify? | Male, Female, Other, Prefer not to say |
| | guardian_age | Guardian age | How old are you? | Under 20, 20–29, 30–39, 40–49, 50–59, 60–69, 70–79, 80+, Prefer not to say |
| | environment | Environment of residence | What type of environment do you and your dog live in? | Rural, Suburban, Urban, Prefer not to say |
| C-BARQ Trainability | cbarq_train_1 | C-BARQ training question 1 | When off the leash, returns immediately when called | Never, Seldom, Sometimes, Usually, Always, Not observed |
| | cbarq_train_2 | C-BARQ training question 2 | Obeys the "sit" command immediately | Never, Seldom, Sometimes, Usually, Always, Not observed |
| | cbarq_train_3 | C-BARQ training question 3 | Obeys the "stay" command immediately | Never, Seldom, Sometimes, Usually, Always, Not observed |
| | cbarq_train_4 | C-BARQ training question 4 | Seems to attend/listen closely to everything you say or do | Never, Seldom, Sometimes, Usually, Always, Not observed |
| | cbarq_train_5 | C-BARQ training question 5 | Slow to respond to correction or punishment | Never, Seldom, Sometimes, Usually, Always, Not observed |
| | cbarq_train_6 | C-BARQ training question 6 | Slow to learn new tricks or tasks | Never, Seldom, Sometimes, Usually, Always, Not observed |

(Contd.)

| CATEGORY OF VARIABLE | VARIABLE NAME | DESCRIPTION | QUESTION TEXT | POSSIBLE RESPONSE VALUES |
|---|---|---|---|---|
| | cbarq_train_7 | C-BARQ training question 7 | Easily distracted by interesting sights, sounds, or smells | Never, Seldom, Sometimes, Usually, Always, Not observed |
| | cbarq_train_8 | C-BARQ training question 8 | Will "fetch," or attempt to fetch, sticks, balls, or objects | Never, Seldom, Sometimes, Usually, Always, Not observed |
| Opt-Out Point | continue_cbarq | Status of whether owner continued to remaining C-BARQ questions | Thank you so much for your answers! At this point in the survey, you have completed the minimum amount required to participate in ManyDogs Study 1, and can choose to submit your information now by selecting 'Submit my info now'. If you would like to tell us more about your dog, we would love to hear all about them! We have prepared several more questions about their behaviour that you can answer by selecting 'More questions please', this will take approximately 12–15 minutes. | Yes (Continue to take full C-BARQ), No (Decline to complete full C-BARQ) |
| C-BARQ Aggression | cbarq_aggression_1 | C-BARQ aggression question 1 ("Some dogs display aggressive behavior from time to time.") | When verbally corrected or punished (scolded, shouted at, etc) by you or a household member. | No aggression (No visible signs of aggression), Mild aggression, Moderate aggression (Growling/barking/baring teeth), High aggression, Serious aggression (Snaps, bites, or attempts to bite), Not observed |
| | cbarq_aggression_2 | C-BARQ aggression question 2 ("Some dogs display aggressive behavior from time to time.") | When approached directly by an unfamiliar adult while being walked/exercised on a leash | No aggression (No visible signs of aggression), Mild aggression, Moderate aggression (Growling/barking/baring teeth), High aggression, Serious aggression (Snaps, bites, or attempts to bite), Not observed |
| | cbarq_aggression_3 | C-BARQ aggression question 3 ("Some dogs display aggressive behavior from time to time.") | When approached directly by an unfamiliar child while being walked/exercised on a leash | No aggression (No visible signs of aggression), Mild aggression, Moderate aggression (Growling/barking/baring teeth), High aggression, Serious aggression (Snaps, bites, or attempts to bite), Not observed |
| | cbarq_aggression_4 | C-BARQ aggression question 4 ("Some dogs display aggressive behavior from time to time.") | Toward unfamiliar persons approaching the dog while s/he is in your car (at the gas station for example). | No aggression (No visible signs of aggression), Mild aggression, Moderate aggression (Growling/barking/baring teeth), High aggression, Serious aggression (Snaps, bites, or attempts to bite), Not observed |
| | cbarq_aggression_5 | C-BARQ aggression question 5 ("Some dogs display aggressive behavior from time to time.") | When toys, bones or other objects are taken away by a household member | No aggression (No visible signs of aggression), Mild aggression, Moderate aggression (Growling/barking/baring teeth), High aggression, Serious aggression (Snaps, bites, or attempts to bite), Not observed |
| | cbarq_aggression_6 | C-BARQ aggression question 6 ("Some dogs display aggressive behavior from time to time.") | When bathed or groomed by a household member | No aggression (No visible signs of aggression), Mild aggression, Moderate aggression (Growling/barking/baring teeth), High aggression, Serious aggression (Snaps, bites, or attempts to bite), Not observed |

(Contd.)

ManyDogs Project et al. *Journal of Open Psychology Data* DOI: 10.5334/jopd.109

| CATEGORY OF VARIABLE | VARIABLE NAME | DESCRIPTION | QUESTION TEXT | POSSIBLE RESPONSE VALUES |
|---|---|---|---|---|
| | cbarq_aggression_7 | C-BARQ aggression question 7 ("Some dogs display aggressive behavior from time to time.") | When an unfamiliar person approaches you or another member of your family at home. | No aggression (No visible signs of aggression), Mild aggression, Moderate aggression (Growling/barking/baring teeth), High aggression, Serious aggression (Snaps, bites, or attempts to bite), Not observed |
| | cbarq_aggression_8 | C-BARQ aggression question 8 ("Some dogs display aggressive behavior from time to time.") | When unfamiliar persons approach you or another member of your family away from home. | No aggression (No visible signs of aggression), Mild aggression, Moderate aggression (Growling/barking/baring teeth), High aggression, Serious aggression (Snaps, bites, or attempts to bite), Not observed |
| | cbarq_aggression_9 | C-BARQ aggression question 9 ("Some dogs display aggressive behavior from time to time.") | When approached directly by a household member while s/he (the dog) is eating | No aggression (No visible signs of aggression), Mild aggression, Moderate aggression (Growling/barking/baring teeth), High aggression, Serious aggression (Snaps, bites, or attempts to bite), Not observed |
| | cbarq_aggression_10 | C-BARQ aggression question 10 ("Some dogs display aggressive behavior from time to time.") | When mailmen or other delivery workers approach your home. | No aggression (No visible signs of aggression), Mild aggression, Moderate aggression (Growling/barking/baring teeth), High aggression, Serious aggression (Snaps, bites, or attempts to bite), Not observed |
| | cbarq_aggression_11 | C-BARQ aggression question 11 ("Some dogs display aggressive behavior from time to time.") | When his/her food is taken away by a household member. | No aggression (No visible signs of aggression), Mild aggression, Moderate aggression (Growling/barking/baring teeth), High aggression, Serious aggression (Snaps, bites, or attempts to bite), Not observed |
| | cbarq_aggression_12 | C-BARQ aggression question 12 ("Some dogs display aggressive behavior from time to time.") | When strangers walk past your home while your dog is outside or in the yard. | No aggression (No visible signs of aggression), Mild aggression, Moderate aggression (Growling/barking/baring teeth), High aggression, Serious aggression (Snaps, bites, or attempts to bite), Not observed |
| | cbarq_aggression_13 | C-BARQ aggression question 13 ("Some dogs display aggressive behavior from time to time.") | When an unfamiliar person tries to touch or pet the dog. | No aggression (No visible signs of aggression), Mild aggression, Moderate aggression (Growling/barking/baring teeth), High aggression, Serious aggression (Snaps, bites, or attempts to bite), Not observed |
| | cbarq_aggression_14 | C-BARQ aggression question 14 ("Some dogs display aggressive behavior from time to time.") | When joggers, cyclists, rollerbladers or skateboarders pass your home while your dog is outside or in the yard. | No aggression (No visible signs of aggression), Mild aggression, Moderate aggression (Growling/barking/baring teeth), High aggression, Serious aggression (Snaps, bites, or attempts to bite), Not observed |
| | cbarq_aggression_15 | C-BARQ aggression question 15 ("Some dogs display aggressive behavior from time to time.") | When approached directly by an unfamiliar male dog while being walked/exercised on a leash | No aggression (No visible signs of aggression), Mild aggression, Moderate aggression (Growling/barking/baring teeth), High aggression, Serious aggression (Snaps, bites, or attempts to bite), Not observed |
| | cbarq_aggression_16 | C-BARQ aggression question 16 ("Some dogs display aggressive behavior from time to time.") | When approached directly by an unfamiliar female dog while being walked/exercised on a leash | No aggression (No visible signs of aggression), Mild aggression, Moderate aggression (Growling/barking/baring teeth), High aggression, Serious aggression (Snaps, bites, or attempts to bite), Not observed |

| CATEGORY OF VARIABLE | VARIABLE NAME | DESCRIPTION | QUESTION TEXT | POSSIBLE RESPONSE VALUES |
|---|---|---|---|---|
| | cbarq_aggression_17 | C-BARQ aggression question 17 ("Some dogs display aggressive behavior from time to time.") | When stared at directly by a member of the household. | No aggression (No visible signs of aggression), Mild aggression, Moderate aggression (Growling/barking/baring teeth), High aggression, Serious aggression (Snaps, bites, or attempts to bite), Not observed |
| | cbarq_aggression_18 | C-BARQ aggression question 18 ("Some dogs display aggressive behavior from time to time.") | Toward unfamiliar dogs visiting your home. | No aggression (No visible signs of aggression), Mild aggression, Moderate aggression (Growling/barking/baring teeth), High aggression, Serious aggression (Snaps, bites, or attempts to bite), Not observed |
| | cbarq_aggression_19 | C-BARQ aggression question 19 ("Some dogs display aggressive behavior from time to time.") | Toward cats, squirrels or other small animals entering your yard. | No aggression (No visible signs of aggression), Mild aggression, Moderate aggression (Growling/barking/baring teeth), High aggression, Serious aggression (Snaps, bites, or attempts to bite), Not observed |
| | cbarq_aggression_20 | C-BARQ aggression question 20 ("Some dogs display aggressive behavior from time to time.") | Toward unfamiliar persons visiting your home. | No aggression (No visible signs of aggression), Mild aggression, Moderate aggression (Growling/barking/baring teeth), High aggression, Serious aggression (Snaps, bites, or attempts to bite), Not observed |
| | cbarq_aggression_21 | C-BARQ aggression question 21 ("Some dogs display aggressive behavior from time to time.") | When barked, growled, or lunged at by another (unfamiliar) dog. | No aggression (No visible signs of aggression), Mild aggression, Moderate aggression (Growling/barking/baring teeth), High aggression, Serious aggression (Snaps, bites, or attempts to bite), Not observed |
| | cbarq_aggression_22 | C-BARQ aggression question 22 ("Some dogs display aggressive behavior from time to time.") | When stepped over by a member of the household. | No aggression (No visible signs of aggression), Mild aggression, Moderate aggression (Growling/barking/baring teeth), High aggression, Serious aggression (Snaps, bites, or attempts to bite), Not observed |
| | cbarq_aggression_23 | C-BARQ aggression question 23 ("Some dogs display aggressive behavior from time to time.") | When you or a household member retrieves food or objects stolen by the dog. | No aggression (No visible signs of aggression), Mild aggression, Moderate aggression (Growling/barking/baring teeth), High aggression, Serious aggression (Snaps, bites, or attempts to bite), Not observed |
| | cbarq_aggression_24 | C-BARQ aggression question 24 ("Some dogs display aggressive behavior from time to time.") | Towards another (familiar) dog in your household (leave blank if no other dogs). | No aggression (No visible signs of aggression), Mild aggression, Moderate aggression (Growling/barking/baring teeth), High aggression, Serious aggression (Snaps, bites, or attempts to bite), Not observed |
| | cbarq_aggression_25 | C-BARQ aggression question 25 ("Some dogs display aggressive behavior from time to time.") | When approached at a favorite resting/sleeping place by another (familiar) household dog (leave blank if no other dogs). | No aggression (No visible signs of aggression), Mild aggression, Moderate aggression (Growling/barking/baring teeth), High aggression, Serious aggression (Snaps, bites, or attempts to bite), Not observed |
| | cbarq_aggression_26 | C-BARQ aggression question 26 ("Some dogs display aggressive behavior from time to time.") | When approached while eating by another (familiar) household dog (leave blank if no other dogs). | No aggression (No visible signs of aggression), Mild aggression, Moderate aggression (Growling/barking/baring teeth), High aggression, Serious aggression (Snaps, bites, or attempts to bite), Not observed |

| CATEGORY OF VARIABLE | VARIABLE NAME | DESCRIPTION | QUESTION TEXT | POSSIBLE RESPONSE VALUES |
|---|---|---|---|---|
| | cbarq_aggression_27 | C-BARQ aggression question 27 ("Some dogs display aggressive behavior from time to time.") | When approached while playing with/chewing a favorite toy, bone, object, etc., by another (familiar) household dog (leave blank if no other dogs). | No aggression (No visible signs of aggression), Mild aggression, Moderate aggression (Growling/barking/baring teeth), High aggression, Serious aggression (Snaps, bites, or attempts to bite), Not observed |
| C-BARQ Fear | cbarq_fear_1 | C-BARQ fear question 1 ("Dogs sometimes show signs of anxiety or fear when exposed to particular sounds, objects, persons or situations") | When approached directly by an unfamiliar adult while away from your home | No fear (No visible signs of fear), Mild fear, Moderate fear (Moderate fear/anxiety), High fear, Extreme fear (Cowers, retreats, or hides), Not observed |
| | cbarq_fear_2 | C-BARQ fear question 2 ("Dogs sometimes show signs of anxiety or fear when exposed to particular sounds, objects, persons or situations") | When approached directly by an unfamiliar child while away from your home | No fear (No visible signs of fear), Mild fear, Moderate fear (Moderate fear/anxiety), High fear, Extreme fear (Cowers, retreats, or hides), Not observed |
| | cbarq_fear_3 | C-BARQ fear question 3 ("Dogs sometimes show signs of anxiety or fear when exposed to particular sounds, objects, persons or situations") | In response to sudden or loud noises (e.g. vacuum cleaner, car backfire, road drills, objects being dropped, etc.) | No fear (No visible signs of fear), Mild fear, Moderate fear (Moderate fear/anxiety), High fear, Extreme fear (Cowers, retreats, or hides), Not observed |
| | cbarq_fear_4 | C-BARQ fear question 4 ("Dogs sometimes show signs of anxiety or fear when exposed to particular sounds, objects, persons or situations") | When unfamiliar persons visit your home | No fear (No visible signs of fear), Mild fear, Moderate fear (Moderate fear/anxiety), High fear, Extreme fear (Cowers, retreats, or hides), Not observed |
| | cbarq_fear_5 | C-BARQ fear question 5 ("Dogs sometimes show signs of anxiety or fear when exposed to particular sounds, objects, persons or situations") | When an unfamiliar person tries to touch or pet the dog. | No fear (No visible signs of fear), Mild fear, Moderate fear (Moderate fear/anxiety), High fear, Extreme fear (Cowers, retreats, or hides), Not observed |
| | cbarq_fear_6 | C-BARQ fear question 6 ("Dogs sometimes show signs of anxiety or fear when exposed to particular sounds, objects, persons or situations") | In heavy traffic | No fear (No visible signs of fear), Mild fear, Moderate fear (Moderate fear/anxiety), High fear, Extreme fear (Cowers, retreats, or hides), Not observed |
| | cbarq_fear_7 | C-BARQ fear question 7 ("Dogs sometimes show signs of anxiety or fear when exposed to particular sounds, objects, persons or situations") | In response to strange or unfamiliar objects on or near the sidewalk (e.g. plastic trash bags, leaves, litter, flags flapping, etc.) | No fear (No visible signs of fear), Mild fear, Moderate fear (Moderate fear/anxiety), High fear, Extreme fear (Cowers, retreats, or hides), Not observed |

(Contd.)

| CATEGORY OF VARIABLE | VARIABLE NAME | DESCRIPTION | QUESTION TEXT | POSSIBLE RESPONSE VALUES |
|---|---|---|---|---|
| | cbarq_fear_8 | C-BARQ fear question 8 ("Dogs sometimes show signs of anxiety or fear when exposed to particular sounds, objects, persons or situations") | When examined/treated by a veterinarian. | No fear (No visible signs of fear), Mild fear, Moderate fear (Moderate fear/anxiety), High fear, Extreme fear (Cowers, retreats, or hides), Not observed |
| | cbarq_fear_9 | C-BARQ fear question 9 ("Dogs sometimes show signs of anxiety or fear when exposed to particular sounds, objects, persons or situations") | During thunderstorms, firework displays, or similar events. | No fear (No visible signs of fear), Mild fear, Moderate fear (Moderate fear/anxiety), High fear, Extreme fear (Cowers, retreats, or hides), Not observed |
| | cbarq_fear_10 | C-BARQ fear question 10 ("Dogs sometimes show signs of anxiety or fear when exposed to particular sounds, objects, persons or situations") | When approached directly by an unfamiliar dog of the same or larger size. | No fear (No visible signs of fear), Mild fear, Moderate fear (Moderate fear/anxiety), High fear, Extreme fear (Cowers, retreats, or hides), Not observed |
| | cbarq_fear_11 | C-BARQ fear question 11 ("Dogs sometimes show signs of anxiety or fear when exposed to particular sounds, objects, persons or situations") | When approached directly by an unfamiliar dog of a smaller size. | No fear (No visible signs of fear), Mild fear, Moderate fear (Moderate fear/anxiety), High fear, Extreme fear (Cowers, retreats, or hides), Not observed |
| | cbarq_fear_12 | C-BARQ fear question 12 ("Dogs sometimes show signs of anxiety or fear when exposed to particular sounds, objects, persons or situations") | When first exposed to unfamiliar situations (e.g. first car trip, first time in elevator, first visit to veterinarian, etc.) | No fear (No visible signs of fear), Mild fear, Moderate fear (Moderate fear/anxiety), High fear, Extreme fear (Cowers, retreats, or hides), Not observed |
| | cbarq_fear_13 | C-BARQ fear question 13 ("Dogs sometimes show signs of anxiety or fear when exposed to particular sounds, objects, persons or situations") | In response to wind or wind-blown objects. | No fear (No visible signs of fear), Mild fear, Moderate fear (Moderate fear/anxiety), High fear, Extreme fear (Cowers, retreats, or hides), Not observed |
| | cbarq_fear_14 | C-BARQ fear question 14 ("Dogs sometimes show signs of anxiety or fear when exposed to particular sounds, objects, persons or situations") | When having nails clipped by a household member. | No fear (No visible signs of fear), Mild fear, Moderate fear (Moderate fear/anxiety), High fear, Extreme fear (Cowers, retreats, or hides), Not observed |
| | cbarq_fear_15 | C-BARQ fear question 15 ("Dogs sometimes show signs of anxiety or fear when exposed to particular sounds, objects, persons or situations") | When groomed or bathed by a household member. | No fear (No visible signs of fear), Mild fear, Moderate fear (Moderate fear/anxiety), High fear, Extreme fear (Cowers, retreats, or hides), Not observed |

(Contd.)

| CATEGORY OF VARIABLE | VARIABLE NAME | DESCRIPTION | QUESTION TEXT | POSSIBLE RESPONSE VALUES |
|---|---|---|---|---|
| | cbarq_fear_16 | C-BARQ fear question 16 ("Dogs sometimes show signs of anxiety or fear when exposed to particular sounds, objects, persons or situations") | When having his/her feet toweled by a member of the household. | No fear (No visible signs of fear), Mild fear, Moderate fear (Moderate fear/anxiety), High fear, Extreme fear (Cowers, retreats, or hides), Not observed |
| | cbarq_fear_17 | C-BARQ fear question 17 ("Dogs sometimes show signs of anxiety or fear when exposed to particular sounds, objects, persons or situations") | When unfamiliar dogs visit your home | No fear (No visible signs of fear), Mild fear, Moderate fear (Moderate fear/anxiety), High fear, Extreme fear (Cowers, retreats, or hides), Not observed |
| | cbarq_fear_18 | C-BARQ fear question 18 ("Dogs sometimes show signs of anxiety or fear when exposed to particular sounds, objects, persons or situations") | When barked, growled, or lunged at by an unfamiliar dog. | No fear (No visible signs of fear), Mild fear, Moderate fear (Moderate fear/anxiety), High fear, Extreme fear (Cowers, retreats, or hides), Not observed |
| C-BARQ Separation-Related Behavior | cbarq_separation_1 | C-BARQ separation question 1 ("Some dogs show signs of anxiety or abnormal behavior when left alone, even for relatively short periods of time.") | Shaking, shivering, or trembling | Never, Seldom, Sometimes, Usually, Always, Not observed |
| | cbarq_separation_2 | C-BARQ separation question 2 ("Some dogs show signs of anxiety or abnormal behavior when left alone, even for relatively short periods of time.") | Excessive Salivation | Never, Seldom, Sometimes, Usually, Always, Not observed |
| | cbarq_separation_3 | C-BARQ separation question 3 ("Some dogs show signs of anxiety or abnormal behavior when left alone, even for relatively short periods of time.") | Restlessness/agitation/pacing | Never, Seldom, Sometimes, Usually, Always, Not observed |
| | cbarq_separation_4 | C-BARQ separation question 4 ("Some dogs show signs of anxiety or abnormal behavior when left alone, even for relatively short periods of time.") | Whining | Never, Seldom, Sometimes, Usually, Always, Not observed |
| | cbarq_separation_5 | C-BARQ separation question 5 ("Some dogs show signs of anxiety or abnormal behavior when left alone, even for relatively short periods of time.") | Barking | Never, Seldom, Sometimes, Usually, Always, Not observed |

(Contd.)

| CATEGORY OF VARIABLE | VARIABLE NAME | DESCRIPTION | QUESTION TEXT | POSSIBLE RESPONSE VALUES |
|---|---|---|---|---|
| | cbarq_separation_6 | C-BARQ separation question 6 ("Some dogs show signs of anxiety or abnormal behavior when left alone, even for relatively short periods of time.") | Howling | Never, Seldom, Sometimes, Usually, Always, Not observed |
| | cbarq_separation_7 | C-BARQ separation question 7 ("Some dogs show signs of anxiety or abnormal behavior when left alone, even for relatively short periods of time.") | Chewing/scratching at doors, floor, windows, curtains, etc | Never, Seldom, Sometimes, Usually, Always, Not observed |
| | cbarq_separation_8 | C-BARQ separation question 8 ("Some dogs show signs of anxiety or abnormal behavior when left alone, even for relatively short periods of time.") | Loss of appetite | Never, Seldom, Sometimes, Usually, Always, Not observed |
| C-BARQ Excitability | cbarq_excitability_1 | C-BARQ excitability question 1 ("Some dogs show relatively little reaction to sudden or potentially exciting events and disturbances in their environment, while others become highly excited at the slightest novelty.") | When you or other members of the household come home after a brief absence. | No excitability (Little or no special reaction), Mild excitability, Moderate excitability, High excitability, Extreme excitability (Over-reacts, hard to calm down), Not observed |
| | cbarq_excitability_2 | C-BARQ excitability question 2 ("Some dogs show relatively little reaction to sudden or potentially exciting events and disturbances in their environment, while others become highly excited at the slightest novelty.") | When playing with you or other members of your household. | No excitability (Little or no special reaction), Mild excitability, Moderate excitability, High excitability, Extreme excitability (Over-reacts, hard to calm down), Not observed |
| | cbarq_excitability_3 | C-BARQ excitability question 3 ("Some dogs show relatively little reaction to sudden or potentially exciting events and disturbances in their environment, while others become highly excited at the slightest novelty.") | When the doorbell rings. | No excitability (Little or no special reaction), Mild excitability, Moderate excitability, High excitability, Extreme excitability (Over-reacts, hard to calm down), Not observed |
| | cbarq_excitability_4 | C-BARQ excitability question 4 ("Some dogs show relatively little reaction to sudden or potentially exciting events and disturbances in their environment, while others become highly excited at the slightest novelty.") | Just before being taken for a walk | No excitability (Little or no special reaction), Mild excitability, Moderate excitability, High excitability, Extreme excitability (Over-reacts, hard to calm down), Not observed |

| CATEGORY OF VARIABLE | VARIABLE NAME | DESCRIPTION | QUESTION TEXT | POSSIBLE RESPONSE VALUES |
|---|---|---|---|---|
| | cbarq_excitability_5 | C-BARQ excitability question 5 ("Some dogs show relatively little reaction to sudden or potentially exciting events and disturbances in their environment, while others become highly excited at the slightest novelty.") | Just before being taken on a car trip | No excitability (Little or no special reaction), Mild excitability, Moderate excitability, High excitability, Extreme excitability (Over-reacts, hard to calm down), Not observed |
| | cbarq_excitability_6 | C-BARQ excitability question 6 ("Some dogs show relatively little reaction to sudden or potentially exciting events and disturbances in their environment, while others become highly excited at the slightest novelty.") | When visitors arrive at your home. | No excitability (Little or no special reaction), Mild excitability, Moderate excitability, High excitability, Extreme excitability (Over-reacts, hard to calm down), Not observed |
| C-BARQ Attachment/ Attention-Seeking | cbarq_attachment_1 | C-BARQ attachment question 1 ("Most dogs are strongly attached to their people, and some demand a great deal of attention and affection from them.") | Displays a strong attachment for one particular member of the household | Never, Seldom, Sometimes, Usually, Always, Not observed |
| | cbarq_attachment_2 | C-BARQ attachment question 2 ("Most dogs are strongly attached to their people, and some demand a great deal of attention and affection from them.") | Tends to follow you (or other members of household) about the house, from room to room | Never, Seldom, Sometimes, Usually, Always, Not observed |
| | cbarq_attachment_3 | C-BARQ attachment question 3 ("Most dogs are strongly attached to their people, and some demand a great deal of attention and affection from them.") | Tends to sit close to, or in contact with, you (or others) when you are sitting down | Never, Seldom, Sometimes, Usually, Always, Not observed |
| | cbarq_attachment_4 | C-BARQ attachment question 4 ("Most dogs are strongly attached to their people, and some demand a great deal of attention and affection from them.") | Tends to nudge, nuzzle or paw you (or others) for attention when you are sitting down | Never, Seldom, Sometimes, Usually, Always, Not observed |
| | cbarq_attachment_5 | C-BARQ attachment question 5 ("Most dogs are strongly attached to their people, and some demand a great deal of attention and affection from them.") | Becomes agitated (whines, jumps up, tries to intervene) when you (or others) show affection for another person | Never, Seldom, Sometimes, Usually, Always, Not observed |

| CATEGORY OF VARIABLE | VARIABLE NAME | DESCRIPTION | QUESTION TEXT | POSSIBLE RESPONSE VALUES |
|---|---|---|---|---|
| | `cbarq_attachment_6` | C-BARQ attachment question 6 ("Most dogs are strongly attached to their people, and some demand a great deal of attention and affection from them.") | Becomes agitated (whines, jumps up, tries to intervene) when you show affection for another dog or animal | Never, Seldom, Sometimes, Usually, Always, Not observed |
| C-BARQ Miscellaneous Behavior Problems | `cbarq_miscellaneous_1` | C-BARQ miscellaneous question 1 ("Dogs display a wide range of miscellaneous behavior problems in addition to those already covered by this questionnaire.") | Chases or would chase cats given the opportunity | Never, Seldom, Sometimes, Usually, Always, Not observed |
| | `cbarq_miscellaneous_2` | C-BARQ miscellaneous question 2 ("Dogs display a wide range of miscellaneous behavior problems in addition to those already covered by this questionnaire.") | Chases or would chase birds given the opportunity | Never, Seldom, Sometimes, Usually, Always, Not observed |
| | `cbarq_miscellaneous_3` | C-BARQ miscellaneous question 3 ("Dogs display a wide range of miscellaneous behavior problems in addition to those already covered by this questionnaire.") | Chases or would chase squirrels, rabbits and other small animals given the opportunity | Never, Seldom, Sometimes, Usually, Always, Not observed |
| | `cbarq_miscellaneous_4` | C-BARQ miscellaneous question 4 ("Dogs display a wide range of miscellaneous behavior problems in addition to those already covered by this questionnaire.") | Escapes or would escape from home or yard given the chance | Never, Seldom, Sometimes, Usually, Always, Not observed |
| | `cbarq_miscellaneous_5` | C-BARQ miscellaneous question 5 ("Dogs display a wide range of miscellaneous behavior problems in addition to those already covered by this questionnaire.") | Rolls in animal droppings or other 'smelly' substances | Never, Seldom, Sometimes, Usually, Always, Not observed |
| | `cbarq_miscellaneous_6` | C-BARQ miscellaneous question 6 ("Dogs display a wide range of miscellaneous behavior problems in addition to those already covered by this questionnaire.") | Eats own or other animals' droppings or feces | Never, Seldom, Sometimes, Usually, Always, Not observed |
| | `cbarq_miscellaneous_7` | C-BARQ miscellaneous question 7 ("Dogs display a wide range of miscellaneous behavior problems in addition to those already covered by this questionnaire.") | Chews inappropriate objects | Never, Seldom, Sometimes, Usually, Always, Not observed |

ManyDogs Project et al. *Journal of Open Psychology Data* DOI: 10.5334/jopd.109

| CATEGORY OF VARIABLE | VARIABLE NAME | DESCRIPTION | QUESTION TEXT | POSSIBLE RESPONSE VALUES |
|---|---|---|---|---|
| | cbarq_miscellaneous_8 | C-BARQ miscellaneous question 8 ("Dogs display a wide range of miscellaneous behavior problems in addition to those already covered by this questionnaire.") | Mounts' objects, furniture, or people | Never, Seldom, Sometimes, Usually, Always, Not observed |
| | cbarq_miscellaneous_9 | C-BARQ miscellaneous question 9 ("Dogs display a wide range of miscellaneous behavior problems in addition to those already covered by this questionnaire.") | Begs persistently for food when people are eating | Never, Seldom, Sometimes, Usually, Always, Not observed |
| | cbarq_miscellaneous_10 | C-BARQ miscellaneous question 10 ("Dogs display a wide range of miscellaneous behavior problems in addition to those already covered by this questionnaire.") | Steals food | Never, Seldom, Sometimes, Usually, Always, Not observed |
| | cbarq_miscellaneous_11 | C-BARQ miscellaneous question 11 ("Dogs display a wide range of miscellaneous behavior problems in addition to those already covered by this questionnaire.") | Nervous or frightened on stairs | Never, Seldom, Sometimes, Usually, Always, Not observed |
| | cbarq_miscellaneous_12 | C-BARQ miscellaneous question 12 ("Dogs display a wide range of miscellaneous behavior problems in addition to those already covered by this questionnaire.") | Pulls excessively hard when on the leash | Never, Seldom, Sometimes, Usually, Always, Not observed |
| | cbarq_miscellaneous_13 | C-BARQ miscellaneous question 13 ("Dogs display a wide range of miscellaneous behavior problems in addition to those already covered by this questionnaire.") | Urinates against objects/ furnishings in your home | Never, Seldom, Sometimes, Usually, Always, Not observed |
| | cbarq_miscellaneous_14 | C-BARQ miscellaneous question 14 ("Dogs display a wide range of miscellaneous behavior problems in addition to those already covered by this questionnaire.") | Urinates when approached, petted, handled or picked up | Never, Seldom, Sometimes, Usually, Always, Not observed |
| | cbarq_miscellaneous_15 | C-BARQ miscellaneous question 15 ("Dogs display a wide range of miscellaneous behavior problems in addition to those already covered by this questionnaire.") | Urinates when left alone at night, or during the daytime | Never, Seldom, Sometimes, Usually, Always, Not observed |

| CATEGORY OF VARIABLE | VARIABLE NAME | DESCRIPTION | QUESTION TEXT | POSSIBLE RESPONSE VALUES |
|---|---|---|---|---|
| | cbarq_miscellaneous_16 | C-BARQ miscellaneous question 16 ("Dogs display a wide range of miscellaneous behavior problems in addition to those already covered by this questionnaire.") | Defecates when left alone at night, or during the daytime | Never, Seldom, Sometimes, Usually, Always, Not observed |
| | cbarq_miscellaneous_17 | C-BARQ miscellaneous question 17 ("Dogs display a wide range of miscellaneous behavior problems in addition to those already covered by this questionnaire.") | Hyperactive, restless, has trouble settling down | Never, Seldom, Sometimes, Usually, Always, Not observed |
| | cbarq_miscellaneous_18 | C-BARQ miscellaneous question 18 ("Dogs display a wide range of miscellaneous behavior problems in addition to those already covered by this questionnaire.") | Playful, puppyish, boisterous | Never, Seldom, Sometimes, Usually, Always, Not observed |
| | cbarq_miscellaneous_19 | C-BARQ miscellaneous question 19 ("Dogs display a wide range of miscellaneous behavior problems in addition to those already covered by this questionnaire.") | Active, energetic, always on the go | Never, Seldom, Sometimes, Usually, Always, Not observed |
| | cbarq_miscellaneous_20 | C-BARQ miscellaneous question 20 ("Dogs display a wide range of miscellaneous behavior problems in addition to those already covered by this questionnaire.") | Stares intently at nothing visible | Never, Seldom, Sometimes, Usually, Always, Not observed |
| | cbarq_miscellaneous_21 | C-BARQ miscellaneous question 21 ("Dogs display a wide range of miscellaneous behavior problems in addition to those already covered by this questionnaire.") | Snaps at (invisible) flies | Never, Seldom, Sometimes, Usually, Always, Not observed |
| | cbarq_miscellaneous_22 | C-BARQ miscellaneous question 22 ("Dogs display a wide range of miscellaneous behavior problems in addition to those already covered by this questionnaire.") | Chases own tail/hind end | Never, Seldom, Sometimes, Usually, Always, Not observed |
| | cbarq_miscellaneous_23 | C-BARQ miscellaneous question 23 ("Dogs display a wide range of miscellaneous behavior problems in addition to those already covered by this questionnaire.") | Chases/follows shadows, light spots, etc. | Never, Seldom, Sometimes, Usually, Always, Not observed |

| CATEGORY OF VARIABLE | VARIABLE NAME | DESCRIPTION | QUESTION TEXT | POSSIBLE RESPONSE VALUES |
|---|---|---|---|---|
| | cbarq_miscellaneous_24 | C-BARQ miscellaneous question 24 ("Dogs display a wide range of miscellaneous behavior problems in addition to those already covered by this questionnaire.") | Barks persistently when alarmed or excited | Never, Seldom, Sometimes, Usually, Always, Not observed |
| | cbarq_miscellaneous_25 | C-BARQ miscellaneous question 25 ("Dogs display a wide range of miscellaneous behavior problems in addition to those already covered by this questionnaire.") | Licks him/herself excessively | Never, Seldom, Sometimes, Usually, Always, Not observed |
| | cbarq_miscellaneous_26 | C-BARQ miscellaneous question 26 ("Dogs display a wide range of miscellaneous behavior problems in addition to those already covered by this questionnaire.") | Licks people or objects excessively | Never, Seldom, Sometimes, Usually, Always, Not observed |
| | cbarq_miscellaneous_27 | C-BARQ miscellaneous question 27 ("Dogs display a wide range of miscellaneous behavior problems in addition to those already covered by this questionnaire.") | Displays other bizarre, strange, or repetitive behavior(s) | Never, Seldom, Sometimes, Usually, Always, Not observed |
| Behavior Testing | first_condition | Which experimental condition was experienced first | | Nonostensive, Ostensive |
| | onecup_1 | Choice in one-cup warm-up trial 1 | | 1 = chose cup with treat, 0 = did not choose cup |
| | onecup_2 | Choice in one-cup warm-up trial 2 | | 1 = chose cup with treat, 0 = did not choose cup |
| | onecup_3 | Choice in one-cup warm-up trial 3 | | 1 = chose cup with treat, 0 = did not choose cup |
| | onecup_4 | Choice in one-cup warm-up trial 4 | | 1 = chose cup with treat, 0 = did not choose cup |
| | onecup_5 | Choice in one-cup warm-up trial 5 | | 1 = chose cup with treat, 0 = did not choose cup |
| | onecup_6 | Choice in one-cup warm-up trial 6 | | 1 = chose cup with treat, 0 = did not choose cup |
| | onecup_7 | Choice in one-cup warm-up trial 7 | | 1 = chose cup with treat, 0 = did not choose cup |
| | twocup_1 | Choice in two-cup warm-up trial 1 | | 1 = chose cup with treat, 0 = chose cup without treat |
| | twocup_2 | Choice in two-cup warm-up trial 2 | | 1 = chose cup with treat, 0 = chose cup without treat |
| | twocup_3 | Choice in two-cup warm-up trial 3 | | 1 = chose cup with treat, 0 = chose cup without treat |
| | twocup_4 | Choice in two-cup warm-up trial 4 | | 1 = chose cup with treat, 0 = chose cup without treat |
| | twocup_5 | Choice in two-cup warm-up trial 5 | | 1 = chose cup with treat, 0 = chose cup without treat |
| | twocup_6 | Choice in two-cup warm-up trial 6 | | 1 = chose cup with treat, 0 = chose cup without treat |

(Contd.)

| CATEGORY OF VARIABLE | VARIABLE NAME | DESCRIPTION | QUESTION TEXT | POSSIBLE RESPONSE VALUES |
|---|---|---|---|---|
| | twocup_7 | Choice in two-cup warm-up trial 7 | | 1 = chose cup with treat, 0 = chose cup without treat |
| | twocup_8 | Choice in two-cup warm-up trial 8 | | 1 = chose cup with treat, 0 = chose cup without treat |
| | twocup_9 | Choice in two-cup warm-up trial 9 | | 1 = chose cup with treat, 0 = chose cup without treat |
| | twocup_10 | Choice in two-cup warm-up trial 10 | | 1 = chose cup with treat, 0 = chose cup without treat |
| | twocup_11 | Choice in two-cup warm-up trial 11 | | 1 = chose cup with treat, 0 = chose cup without treat |
| | twocup_12 | Choice in two-cup warm-up trial 12 | | 1 = chose cup with treat, 0 = chose cup without treat |
| | twocup_13 | Choice in two-cup warm-up trial 13 | | 1 = chose cup with treat, 0 = chose cup without treat |
| | twocup_14 | Choice in two-cup warm-up trial 14 | | 1 = chose cup with treat, 0 = chose cup without treat |
| | twocup_15 | Choice in two-cup warm-up trial 15 | | 1 = chose cup with treat, 0 = chose cup without treat |
| | twocup_16 | Choice in two-cup warm-up trial 16 | | 1 = chose cup with treat, 0 = chose cup without treat |
| | twocup_17 | Choice in two-cup warm-up trial 17 | | 1 = chose cup with treat, 0 = chose cup without treat |
| | twocup_18 | Choice in two-cup warm-up trial 18 | | 1 = chose cup with treat, 0 = chose cup without treat |
| | twocup_19 | Choice in two-cup warm-up trial 19 | | 1 = chose cup with treat, 0 = chose cup without treat |
| | twocup_20 | Choice in two-cup warm-up trial 20 | | 1 = chose cup with treat, 0 = chose cup without treat |
| | twocup_21 | Choice in two-cup warm-up trial 21 | | 1 = chose cup with treat, 0 = chose cup without treat |
| | twocup_22 | Choice in two-cup warm-up trial 22 | | 1 = chose cup with treat, 0 = chose cup without treat |
| | twocup_23 | Choice in two-cup warm-up trial 23 | | 1 = chose cup with treat, 0 = chose cup without treat |
| | nonostensive_1 | Choice in non-ostensive trial 1 | | 1 = chose cup with treat, 0 = chose cup without treat |
| | nonostensive_2 | Choice in non-ostensive trial 2 | | 1 = chose cup with treat, 0 = chose cup without treat |
| | nonostensive_3 | Choice in non-ostensive trial 3 | | 1 = chose cup with treat, 0 = chose cup without treat |
| | nonostensive_4 | Choice in non-ostensive trial 4 | | 1 = chose cup with treat, 0 = chose cup without treat |
| | nonostensive_5 | Choice in non-ostensive trial 5 | | 1 = chose cup with treat, 0 = chose cup without treat |
| | nonostensive_6 | Choice in non-ostensive trial 6 | | 1 = chose cup with treat, 0 = chose cup without treat |
| | nonostensive_7 | Choice in non-ostensive trial 7 | | 1 = chose cup with treat, 0 = chose cup without treat |
| | nonostensive_8 | Choice in non-ostensive trial 8 | | 1 = chose cup with treat, 0 = chose cup without treat |

| CATEGORY OF VARIABLE | VARIABLE NAME | DESCRIPTION | QUESTION TEXT | POSSIBLE RESPONSE VALUES |
|---|---|---|---|---|
| | nonostensive_9 | Choice in non-ostensive trial 9 | | 1 = chose cup with treat, 0 = chose cup without treat |
| | nonostensive_10 | Choice in non-ostensive trial 10 | | 1 = chose cup with treat, 0 = chose cup without treat |
| | nonostensive_11 | Choice in non-ostensive trial 11 | | 1 = chose cup with treat, 0 = chose cup without treat |
| | nonostensive_12 | Choice in non-ostensive trial 12 | | 1 = chose cup with treat, 0 = chose cup without treat |
| | nonostensive_13 | Choice in non-ostensive trial 13 | | 1 = chose cup with treat, 0 = chose cup without treat |
| | nonostensive_14 | Choice in non-ostensive trial 14 | | 1 = chose cup with treat, 0 = chose cup without treat |
| | ostensive_1 | Choice in ostensive trial 1 | | 1 = chose cup with treat, 0 = chose cup without treat |
| | ostensive_2 | Choice in ostensive trial 2 | | 1 = chose cup with treat, 0 = chose cup without treat |
| | ostensive_3 | Choice in ostensive trial 3 | | 1 = chose cup with treat, 0 = chose cup without treat |
| | ostensive_4 | Choice in ostensive trial 4 | | 1 = chose cup with treat, 0 = chose cup without treat |
| | ostensive_5 | Choice in ostensive trial 5 | | 1 = chose cup with treat, 0 = chose cup without treat |
| | ostensive_6 | Choice in ostensive trial 6 | | 1 = chose cup with treat, 0 = chose cup without treat |
| | ostensive_7 | Choice in ostensive trial 7 | | 1 = chose cup with treat, 0 = chose cup without treat |
| | ostensive_8 | Choice in ostensive trial 8 | | 1 = chose cup with treat, 0 = chose cup without treat |
| | ostensive_9 | Choice in ostensive trial 9 | | 1 = chose cup with treat, 0 = chose cup without treat |
| | ostensive_10 | Choice in ostensive trial 10 | | 1 = chose cup with treat, 0 = chose cup without treat |
| | ostensive_11 | Choice in ostensive trial 11 | | 1 = chose cup with treat, 0 = chose cup without treat |
| | odor_1 | Choice in odor trial 1 | | 1 = chose cup with treat, 0 = chose cup without treat |
| | odor_2 | Choice in odor trial 2 | | 1 = chose cup with treat, 0 = chose cup without treat |
| | odor_3 | Choice in odor trial 3 | | 1 = chose cup with treat, 0 = chose cup without treat |
| | odor_4 | Choice in odor trial 4 | | 1 = chose cup with treat, 0 = chose cup without treat |
| | odor_5 | Choice in odor trial 5 | | 1 = chose cup with treat, 0 = chose cup without treat |
| | odor_6 | Choice in odor trial 6 | | 1 = chose cup with treat, 0 = chose cup without treat |

**Table 2** Data description for complete ManyDogs 1 study data.

### 3.1 REPOSITORY LOCATION
The dataset for this study is available on the Open Science Framework at https://osf.io/7rwpc/ (DOI: 10.17605/osf.io/7rwpc) and on GitHub at https://github.com/ManyDogsProject/md1_data.

### 3.2 OBJECT/FILE NAME
The file name for the dataset is `manydogs_etal_2024_data.csv` and the codebook is `manydogs_etal_2024_codebook.csv`.

### 3.3 DATA TYPE
This dataset includes processed data from the ManyDogs 1 study. We have removed identifiable information, recoded data values for consistency, renamed and reordered columns for clarity, and combined survey data submitted by guardians via Qualtrics and behavioral data submitted by research teams via Qualtrics.

### 3.4 FORMAT NAMES AND VERSIONS
The dataset and codebook are provided in a comma-separated (`.csv`) plain text format. There is one version of the dataset with no anticipated additional versions, as data collection has ended.

### 3.5 LANGUAGE
The variable names and text values are in English. Though data were collected in other languages (Croatian, Hungarian, Italian, Polish, and Spanish), the Qualtrics surveys were coded to save responses in English.

### 3.6 LICENSE
The ManyDogs 1 dataset is available under a CC BY 4.0 license, which allows users to share (copy and redistribute the material in any medium or format for any purpose, even commercially) and adapt (remix, transform, and build upon the material for any purpose, even commercially) this material as long as they give appropriate credit, provide a link to the license, indicate if changes were made, and do not apply legal terms or technological measures that legally restrict others from doing anything the license permits.

### 3.7 LIMITS TO SHARING
The dataset is freely available for download on the Open Science Framework. There are no limits to sharing beyond those described in the license.

### 3.8 PUBLICATION DATE
The dataset was uploaded to the Open Science Framework on 2024-02-06 and updated on 2024-05-02.

### 3.9 FAIR DATA/CODEBOOK
This dataset is *findable* through the persistent identifier on the Open Science Framework (DOI: 10.17605/osf.io/7rwpc), *accessible* through free availability on Open Science Framework and GitHub, *interoperable* by using plain-text CSV data files, and *reusable* with the CC-BY 4.0 license. Metadata are included as codebook here (Table 2) and with the data on Open Science Framework and GitHub.

## (4) REUSE POTENTIAL

The original data from ManyDogs 1 (ManyDogs Project et al., 2023b) focuses on dog responses in the two-alternative object-choice task across warm-up, ostensive, non-ostenstive, and odor control trials. In addition, that dataset includes basic demographics on the dog and guardian, as well as the mean trainability score from the C-BARQ. The current dataset adds information on dog origin and household, dog training experience, guardian communication practices, and the complete C-BARQ profile. The C-BARQ data are quite rich, with sections on training, aggression, fear, separation-related behavior, excitability, attachment and attention seeking, and miscellaneous problem behaviors. Thus, this dataset allows for assessing associations among all of the C-BARQ measures as well as connections to the experimental task data and the other dog and guardian characteristic data.

A key strength of this dataset is its diversity. The data were collected by 20 different research sites in eight countries, allowing the assessment of site effects as well as cultural differences. In addition, while most dogs are kept in private homes, the dataset also includes a subset of dogs kept in group housing at working dog facilities. Finally, breed is included, allowing the exploration of breed differences.

One limitation of this dataset is that, though the C-BARQ training survey questions were compulsory for all guardians, the remaining questions were optional to ease the survey burden. As a result, 512 of the 704 guardians elected to continue on to the optional questions (though not all completed the survey). Importantly, the completion rate varied across research sites, ranging from 24.3 to 100.0%, potentially introducing bias in responses to the optional questions across sites.

Despite these limitations, this dataset provides valuable data on dog point-following behavior in the face of conflicting interpretations in the literature as informative or associative (Wynne et al., 2008; Topál et al., 2009; Kaminski et al., 2012; Kaminski & Nitzschner, 2013; Wobber & Kaminski, 2011). Moreover, it provides critical large-scale data investigating particular methodologies used in these tasks (namely contralateral, momentary pointing), which can result in weaker following behavior in dogs (Lyn et al., 2024). The large sample size and the rich demographic data provides one of the most extensive and diverse researcher-collected datasets on dog behavior and cognition. Our hope is that this dataset

will inspire canine scientists to strive for large sample sizes, work across research sites, and collect thorough demographic data to better characterize dog behavior in a way to improve dog welfare and the dog-human bond.

## ACKNOWLEDGEMENTS

We are grateful to all of the research teams and dog guardians who helped generate these data. We are grateful to James Serpell for allowing us to use the C-BARQ questionnaire.

## FUNDING INFORMATION

We are grateful to the Big Team Science Conference for funding the article processing fee via a grant to JE.

## COMPETING INTERESTS

The authors have no competing interests to declare.

## AUTHOR CONTRIBUTIONS

The authors made the following contributions. Julia Espinosa: Conceptualization, Data curation, Formal analysis, Funding acquisition, Methodology, Project administration, Supervision, Writing – original draft, Writing – review & editing; Elizabeth Hare: Conceptualization, Data curation, Formal analysis, Methodology, Project administration, Software, Validation, Writing – original draft, Writing – review & editing; Daniela Alberghina: Investigation, Validation, Writing – original draft, Writing – review & editing; Brian Perez: Investigation, Validation, Writing – original draft, Writing – review & editing; Jeffrey R. Stevens: Conceptualization, Data curation, Formal analysis, Methodology, Project administration, Software, Supervision, Visualization, Writing – original draft, Writing – review & editing.

For the original ManyDogs 1 study, data were collected by: D. Alberghina., H.E.E. Alway, J.D. Barela, E.E. Bray, S.-E. Byosiere, C.M. Cavalli, L.M. Chaudoir, C. Collins-Pisano, H.J. DeBoer, L.E.L.C. Douglas, S. Dror, M.V. Dzik, B. Ferguson, L. Fisher, H.C. Fitzpatrick, M.S. Freeman, S.N. Frinton, M.K. Glover, J.E.P. Goacher, M. Golańska, M. Hickey, H.-L. Jim, D.M. Kelly, V.A. Kuhlmeier, L. Lassiter, L. Lazarowski, J. Leighton-Birch, K. Maliszewska, V. Marra, L.I. Montgomery, M.S. Murray, E.K. Nelson, L. Ostojić, S.G. Palermo, A.E. Parks Russell, M.H. Pelgrim, S.D. Pellowe, A. Reinholz, L.A. Rial, E.M. Richards, M.A. Ross, L.G. Rothkoff, H.Salomons, J.K. Sanger, A.R. Schirle, S.J. Shearer, J.M. Silverman, A. Sommese, T. Srdoc, H. St. John-Mosse, K. Vékony, Y.A. Worth, L.M.I. Zipperling, B. Żołędziewska, and S.G. Zylberfuden.

## AUTHOR AFFILIATIONS

**ManyDogs Project**

**Julia Espinosa** [ID] orcid.org/0000-0003-0780-2762
Department of Human Evolutionary Biology, Harvard University, Cambridge, MA, USA

**Elizabeth Hare** [ID] orcid.org/0000-0002-3978-2543
Dog Genetics LLC, Astoria, NY, USA

**Daniela Alberghina** [ID] orcid.org/0000-0003-2826-0325
Department of Veterinary Sciences, University of Messina, Messina, Italy

**Bryan Mitchel Perez Valverde** [ID] orcid.org/0000-0003-0319-4608
The Graduate Center, City University of New York, New York City, New York, USA

**Jeffrey R. Stevens** [ID] orcid.org/0000-0003-2375-1360
Department of Psychology, Center for Brain, Biology & Behavior, University of Nebraska-Lincoln, Lincoln, Nebraska, USA

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

## PEER REVIEW COMMENTS

Journal of Open Psychology Data has blind peer review, which is unblinded upon article acceptance. The editorial history of this article can be downloaded here:

- **PR File 1.** Peer Review History. DOI: https://doi.org/10.5334/jopd.109.pr1

**TO CITE THIS ARTICLE:**

**Submitted:** 09 February 2024     **Accepted:** 01 August 2024     **Published:** 16 August 2024

