## [Peer Review History. · Journal of Open Psychology Data]

Peer Review

Dear ManyDogs Project, Julia Espinosa, Elizabeth Hare, Daniela Alberghina, Bryan Perez Valverde, Jeffrey Stevens,

After review, we have reached a decision regarding your submission to Journal of Open Psychology Data, "Data from ManyDogs 1". Our decision is to request revisions of the manuscript prior to acceptance for publication.

I think from both reviews and my own reading, this is a really interesting and useful dataset and so to maximise it's impact I think it'd benefit from a little detail and annotation to ensure it can be reused and engaged with as easily as possible. The reviewers have provided some very helpful specific suggestions as to how to increase the clarity and detail of the manuscript and resources and I encourage you to reflect upon these, and the broader additions to clarity you might be able to add, to ensure this work remains accessible and does justice to the massive efforts adopted to collect it!

The full review information is included at the bottom of this email. Please note that there may also be a copy of the manuscript file with reviewer comments available once you have accessed the submission account. We ask you to please consider the comments below and revise the file accordingly.

Instructions for how to resubmit your article online are pasted below. Please ensure that your revised files adhere to our author guidelines, and that the files are fully proofed prior to upload. Please also include a revised version of your article with 'tracked changes', adding comments where appropriate, to indicate the revisions made, in addition to a brief document outlining how you have responded to the reviewers' requests.

If you have trouble processing the revisions, our Help Center (<https://help.u-community.io>) or downloadable PDF (<https://bit.ly/Author-Guide-OJS-3>) may be able to help. If not, please get in touch and we'll be happy to help.

Please also ensure that all copyright permissions have been attained for any figures/tables you have included.

Please could you have the revisions submitted with four weeks. If you cannot make this deadline, please let us know as early as possible.

Kind regards,

Dr Thomas Rhys Evans

Reviewer A:

Recommendation: Revisions Required

Comments to the author(s)

As a reviewer for the manuscript detailing the ManyDogs 1 project, I commend the authors for their ambitious undertaking and the meticulous effort to standardize a complex, multi-site experimental design. The concept of leveraging 'big team science' to tackle questions of canine cognition, specifically dogs' perception of human pointing gestures, represents a significant contribution to the field. The study's scope, involving multiple countries and a large sample size, provides a valuable dataset with the potential to advance our understanding of interspecies communication. Additionally, the commitment to open science, through the sharing of data and methodologies, enhances the study's credibility and utility for future research endeavours.

However, the manuscript would benefit from more explicit details regarding the protocols for data sharing and the process for replicating the study. Enhancing the clarity around these aspects could greatly facilitate subsequent research efforts, enabling deeper and more comprehensive exploration of canine cognition and interspecies communication within the academic community.

- * The manuscript introduces 'big team science' as foundational to its methodology. Could the authors succinctly define 'big team science' and elucidate its significance for this particular study?
- * The experimental setup mentions the use of 'gaze' alongside pointing gestures. Could the authors clarify whether this refers exclusively to human eye gaze directed at the dog and how this interaction of gaze and gesture serves as ostensive cues influencing dog behaviour?
- * The comparison of two styles of pointing is central to the study. Could the authors provide a detailed description of these styles at their first mention in the manuscript to enhance clarity and support the study's reproducibility?
- * To improve the interpretability of Figure 1, could the authors include a legend explaining the significance of the colour coding (e.g., light and dark blues) used within the figure?
- * After the initial C-BARQ trainability questions, guardians could answer additional questions on six behaviour assessment scales. For completeness, could the authors list and briefly describe these scales?
- * The statistical model references a 'correct choice' variable. Could the authors elaborate on the operational definition of 'correct choice' within the study and its integration with other variables in the analysis?
- * Given the multi-site nature of the study, how did the research teams ensure uniform application of the experimental protocols across all sites, especially considering potential cultural and environmental influences on dog behavior?
- * It's noted that not all guardians completed the optional questions of the C-BARQ. How

might the non-responses or partial responses influence the findings, and were any statistical methods used to address potential biases arising from incomplete data sets?

* While the dataset is available under a CC BY 4.0 license, could the authors discuss any anticipated restrictions on data use or specific ways they envision the research community utilizing this dataset?

* Considering the anticipated variance within the dataset, could the authors detail the statistical techniques used to manage this variance? How did this influence the interpretation of the 'correct choice' variable and the alignment of results with preregistered hypotheses?

* Given the distinct predictions for the ostensive vs. non-ostensive conditions, how did the actual performance of dogs across these conditions compare with the preregistered expectations? A specific breakdown of dogs' performance in relation to these predictions would clarify the role of social communicative cues in their behaviour.

* The preregistration details the evaluation of additional predictors such as demographics and training history. Could the authors discuss any notable patterns or effects of these predictors on dogs' performance?

* The preregistration specifies criteria for excluding individual subjects and labs from the main analyses. Based on these criteria, how many subjects and labs were excluded, and what impact, if any, did these exclusions have on the study's overall findings and conclusions?

* The study provides a valuable dataset for understanding dogs' responses to human pointing gestures. How do the authors suggest this data could be integrated with existing literature to advance the field of canine cognition?

* Given the anticipated variance across multiple dimensions of the study, can the authors elaborate on the statistical techniques used to manage and interpret this variance? Specifically, how did they ensure that the conclusions drawn from the data robustly reflect the effects of the experimental conditions over the potential noise introduced by such variability?

* The use of both frequentist statistics and Bayes factors is mentioned. Could the authors discuss the implications of these analytical approaches for interpreting the study's results?

* Finally, based on the findings and data collected, what are the authors' recommendations for future research directions in the field of canine behaviour and cognition?

I hope my feedback is helpful, and I wish you the best in refining this important work.

Reviewer N:

Recommendation: Revisions Required

Comments to the author(s)

The paper presents the dataset for the ManyDogs 1 project. The presentation of the dataset on dogs' pointing involved 704 participants and includes 158 variables.

The manuscript is well written and presents the dataset with clarity. I have minor comments on the manuscript and some suggestions for the presentation of the data within the dataset, mostly concerning definitions of concepts (eg., types of training mentioned, definitions for the C-BARQ measures). Perhaps the addition of a fifth column in the codebook can facilitate the task.

Comments:

Could the authors elaborate more on the following sentence: " Domestic dogs (*Canis familiaris*) have become a popular animal model for investigating behavioral and cognitive evolution due to their shared ecological niche with humans"

What do the authors refer to momentary point? Did the point lasted around a second?

In page 9, I would write "The two EXPERIMENTAL conditions were separated by a one-minute play break and re-familiarization with the testing situation.

Some DOIs do not work (eg <https://doi.org/10.17605/OSF.IO/7RWPC/>)

Once the dogs were acclimated to the room they completed a number of warm-up object-choice tasks in which food was hidden under cups and they had to approach a cup to receive any food rewards hidden underneath. How do these warm-up trials differ from the single cup and two cups warm-up trials at the beginning of the test sessions?

I do not understand why the dataset does not include the trial by trial data—even though it is published elsewhere. Would not be clearer to include all the data here in relation to the other variables? Could the authors give a reason why is better not to include this data?

Dataset

The site column is unclear. Perhaps site information is clearer when looking at Table S1 of ManyDogs Project et al. (2023b). In my opinion, is easier if people can understand all the information in the dataset without assessing other manuscripts. I would add more information on the codebook as possible response values or as meta information in a fifth column in the codebook.

For the training_type column the information is presented in alphabetical order starting with possible response "agility". Why not to continue with the alphabetical order in the subsequent columns? (i.e. training_freq_ability column before training_freq_puppy). In addition, I would define what each type of training usually involve. Some training is evident, while other types of training are rather specific (e.g., Skijoring). For consistence I would define each type of training mentioned.

Not all the readers are familiar with the C-BARQ. I would add the definitions for aggression, separation, fear in the context of the C-BARQ in a fifth column and what each level (mild aggression, mordate, etc) means. This would facilitate the comprehension of the database without assessing other sources of information.

This is just a preference, but I would include the experimental data at the beginning of the task before the training they participated in.